# A Convenient Oligonucleotide Conjugation via Tandem Staudinger Reaction and Amide Bond Formation at the Internucleotidic Phosphate Position

**DOI:** 10.3390/ijms25042007

**Published:** 2024-02-07

**Authors:** Kristina V. Klabenkova, Polina V. Zhdanova, Ekaterina A. Burakova, Sergei N. Bizyaev, Alesya A. Fokina, Dmitry A. Stetsenko

**Affiliations:** 1Department of Physics, Novosibirsk State University, 2 Pirogov Str., Novosibirsk 630090, Russia; k.klabenkova@g.nsu.ru (K.V.K.); burakovaea@bionet.nsc.ru (E.A.B.); bizyaevsn@bionet.nsc.ru (S.N.B.); fokinaaa@bionet.nsc.ru (A.A.F.); 2Institute of Cytology and Genetics, Siberian Branch of the Russian Academy of Sciences, 10 Lavrentiev Ave., Novosibirsk 630090, Russia; 3Department of Natural Sciences, Novosibirsk State University, 2 Pirogov Str., Novosibirsk 630090, Russia; chem13342@gmail.com

**Keywords:** nucleic acid, antisense oligonucleotide, conjugation, sulfonyl azide, pentafluorophenyl, 4-nitrophenyl active esters, carboxylic acid group modification, pentafluorobenzyl

## Abstract

Staudinger reaction on the solid phase between an electronodeficit organic azide, such as sulfonyl azide, and the phosphite triester formed upon phosphoramidite coupling is a convenient method for the chemical modification of oligonucleotides at the internucleotidic phosphate position. In this work, 4-carboxybenzenesulfonyl azide, either with a free carboxy group or in the form of an activated ester such as pentafluorophenyl, 4-nitrophenyl, or pentafluorobenzyl, was used to introduce a carboxylic acid function to the terminal or internal internucleotidic phosphate of an oligonucleotide via the Staudinger reaction. A subsequent treatment with excess primary alkyl amine followed by the usual work-up, after prior activation with a suitable peptide coupling agent such as a uronium salt/1-hydroxybenzotriazole in the case of a free carboxyl, afforded amide-linked oligonucleotide conjugates in good yields including multiple conjugations of up to the exhaustive modification at each phosphate position for a weakly activated pentafluorobenzyl ester, whereas more strongly activated and, thus, more reactive aryl esters provided only single conjugations at the 5′-end. The conjugates synthesized include those with di- and polyamines that introduce a positively charged side chain to potentially assist the intracellular delivery of the oligonucleotide.

## 1. Introduction

The opening decades of this century witnessed an upsurge of interest in synthetic nucleic acids: single- or double-stranded DNA and RNA, and their chemically-modified analogs, which have firmly established their kind as promising therapeutics to engage various target genes and their products at the pre- or post-transcriptional level [1,2]. In comparison to small-molecule drugs, oligonucleotides are capable of recognizing and directly binding the target, which is most often a biologically important RNA molecule, through Watson–Crick complementary interactions, which ensures the high specificity of their action. The attention to oligonucleotide therapy was sparked in the late 1970s with the advent of antisense technology [3], with the subsequent discovery of RNA interference (RNAi) in the late 1990s adding another dimension to it [4]. To date, many oligonucleotide analogs with chemical modifications in various parts of the molecule—nucleobases, sugar, or phosphate backbone—have been developed. The first generation of antisense agents included derivatives with modifications in the phosphate group to ensure enzymatic resistance in the body, such as phosphorothioates [5], methyl phosphonates [6], and many others [7,8,9,10], up to the most recent mesyl phosphoramidates [11,12,13] and phosphoryl guanidines [14,15,16]. The next wave involved compounds with modifications in the ribose ring, such as 2′-*O*-methyl [17,18] or 2′-*O*-methoxyethyl (MOE) RNAs [19,20], 2′-α- or β-fluoro-DNAs [21,22], bridged/locked nucleic acids (B/LNAs) [23,24,25,26], and tricyclo-DNAs [27]. A merger of the above two types are gapmers with 2′-modified 3′ and 5′-terminal “wings” and a 6–10 nt DNA stretch in between, usually all-phosphate-modified [19,20]. A separate group encompasses DNA mimics in which the sugar and phosphate backbone is replaced by an unnatural substitute; examples are peptide nucleic acids (PNAs) [28] and phosphordiamidate morpholino oligomers (PMOs) [29,30]. Up until now, the FDA has approved more than a dozen nucleic-acid-based drugs: notably, nusinersen (Spinraza) [31], and eteplirsen (Exondys 51) [32] and its morpholino kin [33,34], all of which act on rare genetic disease targets through the antisense mechanism. Secondly, there came RNAi-mediating small interfering RNAs (siRNAs) beginning with patisiran (Onpattro) [35], which was quickly followed by others [36,37,38,39]. Furthermore, well over a hundred oligonucleotide drug candidates are currently going through various phases of clinical trials [40].

Thus, there is little doubt that oligonucleotides and their analogs have huge therapeutic potential, which is confirmed every year by the FDA’s approval of yet another nucleic acid drug [41]. However, due to their intrinsically low cellular uptake and tissue distribution, and, in many cases, unfavorable pharmacokinetics and rapid clearance from the body, the full therapeutic effect of nucleic acid drugs has not yet been determined [42,43]. Moreover, the inability of oligonucleotides, more often than not, to effectively overcome additional barriers within the cell on the way to their biological target, such as endosomal entrapment and nuclear translocation [44,45,46,47], poses further obstacles to their use in the clinic.

To improve the therapeutic activity of oligonucleotides, various delivery agents were proposed, such as lipids [48,49,50], polymers [51], particularly polyethyleneimine (PEI) [52], dendrimers [53], inorganic nanoparticles [54], cell-penetrating peptides [55], and oligonucleotide conjugates with different transport molecules [56]. However, despite huge advances recently having been made in this field [57], there is still no generally applicable or inexpensive means of boosting the in vivo efficacy of oligonucleotide therapeutics through improved delivery. A potential step toward the solution of this problem could be to reduce the total negative charge of the phosphate backbone, e.g., through the introduction of cationic moieties, which has previously been shown to improve intracellular penetration of nucleic acids [58,59]. In particular, the attachment of multiple copies of spermine to neutralize a significant proportion of the net negative charge of the oligonucleotide has been shown to promote cellular uptake in the absence of transfection agents [60] as well as to increase the affinity to the nucleic acid target through electrostatic interaction [61,62]. However, the attachment of spermine according to the above method requires an expensive and bulky phosphoramidite synthon [63] whose coupling efficacy, particularly upon multiple incorporations, may be compromised by the generally lower stability of the primary alcohol-derived phosphoramidites compared to, e.g., nucleoside phosphoramidites (especially after storage in a freezer, even at −20 °C). Herein, a post-synthetic conjugation, e.g., via the amide bond formation between a carboxylic acid and a (poly)amine, may offer a useful alternative.

We [64] and others [65] have described phosphoramidite synthons for the modification of synthetic oligonucleotides with a carboxyl group, either in the form of a protected carboxylic ester, which can be cleaved selectively, or as the free carboxyl, which is then activated by a suitable peptide coupling agent either on solid phase [64] or post-synthetically in solution [66] to couple to a range of primary amines, including polyamines and short peptides. An alternative approach involved a phosphoramidite incorporating a pre-activated carboxyl group in the form of *N*-succinimidyl ester that can be converted to a range of amides upon amine treatment in the solid phase [65]. However, this synthon (5′-Carboxy-Modifier C10) as well as the other commercially available reagent (5′-Carboxy-Modifier C5) can only work for the 5′-terminal conjugation, whereas the spermine phosphoramidite offers more flexibility as it can be incorporated on the 5′ or 3′-end, or on both ends at the same time [63]. Some extra flexibility can be achieved by switching to solid-phase conjugation at the 2′-position via 2′-*O*-(carboxymethyl)-uridine synthon, but this approach is sequence-dependent [67]. In order to identify a conjugation method that may be applicable to any position within oligonucleotide chain, we decided to focus on phosphate modification.

The Staudinger reaction has been explored as a way of converting internucleotidic phosphate to the phosphoramidate moiety since as early as the 1970s [68]. In particular, an early paper by Letsinger and Schott described a conjugation of ethidium bromide dye to the internucleotidic position of an oligonucleotide via the Staudinger reaction [69]. Later, Heindl et al. introduced a convenient *N*-(sulfonyl)-phosphoramidate chemistry [70,71], which we recently extended to a range of novel internucleotidic groups [72,73] including, most notably, the mesyl phosphoramidate group [74] that proved to be useful as a backbone modification of antisense oligonucleotides [11,12,13].

Recently, an approach to oligonucleotide functionalization by various reporter groups via sulfonyl azides was published, which presented a further extension of the Staudinger chemistry at the internucleotidic position [75]. Very recently, it was successfully applied to siRNA-peptide conjugation [76]. In this paper, we describe an orthogonal approach to oligonucleotide modification and conjugation at the internucleotidic position via sulfonyl-azide-mediated Staudinger reaction in tandem with amide bond formation between a side-chain carboxyl group and a range of amines including di- and polyamines.

## 2. Results and Discussion

We have previously shown that the Staudinger reaction between an organic azide and the support-bound internucleotidic β-cyanoethyl phosphite triester (**1**, Figure 1), formed upon phosphoramidite coupling during solid-phase DNA synthesis, can serve as a versatile and convenient method for the chemical modification of oligonucleotides at the phosphate position [77]. In this work, we used commercially available 4-carboxybenzenesulfonyl azide **A** or its readily prepared activated esters (**B**–**D**) to modify one or more internucleotidic phosphate groups of an oligonucleotide at the ends or inside the sequence via the Staudinger reaction (**2a**–**d**, Figure 1). Thus, a carboxylic acid function, either free or in its pre-activated form, can be introduced into various positions of an oligonucleotide chain at the phosphate position and subsequently employed to react with a suitable primary amine (see Table 1) with the formation of a stable amide bond. The Staudinger reaction with azides **A**–**D** can be performed manually after the synthesis of the corresponding oligonucleotide at the last phosphate position adjacent the 5′-end, skipping the usual iodine oxidation. More conveniently, it can be conducted automatically by substituting azide solution for standard oxidizer in the synthesizer bottle during solid-phase phosphoramidite DNA synthesis (see Section 4).

Initially, we employed the commercially available 4-carboxybenzenesulfonyl azide **A** to introduce a free carboxy group at the internucleotidic phosphate position next to the 5′-end. A 0.25 M solution of **A** in acetonitrile was used that contained 5% pyridine to buffer the acidity of the carboxylic acid function to avoid any danger of the premature cleavage of the acid-labile 5′-DMTr group and the acid-mediated scission of the phosphite triester. After the Staudinger reaction, an activation of the carboxyl group was carried out on solid phase using a well-known peptide coupling agent, HBTU, in the presence of HOBt [78,79,80] similarly to previously published methods [64,67]. This was followed by a treatment with a solution of a primary amine (Table 1) in acetonitrile at 25 °C for 1 h. At the end of the reaction, cleavage from support and deprotection of the oligonucleotides under mild conditions was ensured by the addition of conc. aq. ammonia solution and standby at 25 °C for 18 h. Following the usual work-up, a series of novel oligonucleotide derivatives containing an *N*-(4-carboxybenzenesulfonyl)-phosphoramidate (ξ) group with different amide-attached side chains were obtained (Table 1). The crude oligonucleotides modified at the 5′-terminal internucleotidic position were usually cleaved retaining the 5′-DMTr group, which can be removed by acidic treatment (80% aq. acetic acid, 30 min) on or after the purification step. The structures of modified oligonucleotide and conjugates were confirmed by MALDI-TOF MS (Appendix A), and the HPLC conversions were good to acceptable (Figure 2A, Appendix A). Thereby, after the usual post-synthetic processing, we obtained a number of amide-linked oligonucleotide conjugates with amines, including, in particular, polycyclic aromatic or polyamine residues, or zwitterionic groups in good yields (Table 1).

Reaction mixtures of oligonucleotide or conjugate syntheses were analyzed by reverse-phased (RP) HPLC. In some profiles such as for a diamine (**5**) or a polyamine (**6**) (Table 1), by-products associated with acrylonitrile addition to the free primary or secondary amino groups were obtained (Appendix A). To avoid this side-reaction, it was found to be advantageous to pre-treat the support-bound oligonucleotide with 50% triethylamine in acetonitrile for 30 min to remove the β-cyanoethyl groups from the internucleotidic phosphates and convert the carboxyl group to its triethylammonium salt to promote the subsequent activation with HBTU/HOBt [81].

Next, we set out to ascertain whether the unprotected carboxyl group remains unchanged during solid-phase DNA synthesis according to the phosphoramidite scheme, by introducing it in the next to the 3′-end or internal position within the oligonucleotide chain. Therefore, the activation and amine treatment steps were skipped, and, after the mild aq. ammonia treatment at 25 °C to avoid potential amide formation, the oligonucleotides were isolated as carboxylic acids (Table 1, **A1**–**A5**). Thus, we demonstrated that the carboxyl group does not need special protection during DNA synthesis and can be introduced into any internucleotidic position within the oligonucleotide chain, including the 3′-terminal or internal position. This may open up further possibilities for conjugation at a modified phosphate group through amide formation upon activation in aqueous or aqueous–organic solution by, e.g., a water-soluble carbodiimide, as described previously [66]. It was possible to clearly distinguish the oligonucleotide with a free carboxyl group (**A3**) from the one with an amide group (**D2**) by co-injection on RP-HPLC (Figure 3A). Paired peaks of the corresponding P-derived diastereomeric oligonucleotides were present.

To determine whether the in situ activation of a free carboxyl group can be skipped using a coupling agent in the solid phase, which may potentially lead to unwanted side-reactions such as base modification [82], we synthesized the activated pentafluorophenyl ester of (4-azidosulfonyl)-benzoic acid **B** through a DCC-mediated reaction of **A** with pentafluorophenol as a stable crystalline solid (see Section 4 for details). It was successfully used to obtain amide-linked oligonucleotide conjugates at the next-to-5′-end internucleotidic position (Table 1, the B series). However, the introduction of the **B** modification at any internal position within the oligonucleotide chain resulted in multiple product formation, indicating that the pentafluorophenyl ester is too reactive to withstand the conditions of solid-phase DNA synthesis. Furthermore, the after-synthesis treatment with 50% triethylamine in acetonitrile to remove the β-cyanoethyl group from the phosphates to avoid potential acrylonitrile addition to the amino groups side-reaction in the case of B5 or B6 was found to be detrimental to the integrity of the active ester. Therefore, a less-activated 4-nitrophenyl (4-azidosulfonyl)-benzoate **C** was obtained to see whether the 4-nitrophenyl group can survive the conditions of DNA synthesis unchanged. However, neither the internally-linked conjugates nor multiple conjugates were available with 4-nitrophenyl ester **C**, and the yields were lower than with **B**. Further attempts to lower the reactivity of aryl ester by making 4-chlorophenyl 4-(azidosulfonyl)-benzoate were likewise unsuccessful to obtain either 3′-terminal or internal modification in acceptable yields. Thus, we tried a still-less-activated pentafluorobenzyl ester of 4-(azidosulfonyl)-benzoic acid **D** to introduce a more stable, yet sufficiently reactive, ester group into the oligonucleotides via the Staudinger reaction. Contrary to both aryl esters **B** and **C**, a relatively weakly activated pentafluorobenzyl ester **D** proved to be stable under the conditions of solid-phase oligonucleotide synthesis. The application of ester **D** for the modification of internucleotidic phosphate groups allowed us to obtain amide-linked conjugates without pre-activation by post-synthetic treatment with conc. aq. ammonia (**1**) or 1,1-dimethylethylenediamine (**7**) (Table 1, Figure 4), including those with next-to-the-3′-terminal or internal modification such as **D1**, **D3**, **D5**/**D7**, or more than one conjugated position, e.g., **D4**, up to the exhaustive replacement of all of the internucleotidic phosphates with modified groups, e.g., **D6**/**D8** (Table 1).

Typical elution profiles of the obtained amide-linked oligonucleotide conjugates are presented in Figure 2, Figure 3, Figure 4 and Figure 5 (see also Appendix A). As expected, the replacement of one phosphate group with *N*-(4-carboxybenzenesulfonyl)-phosphoramidate (ξ) in the model 6-mer hexathymidylate **A1** led to a slight increase in the retention time compared to the unmodified one (Figure 2B). In the conjugate with benzylamine **A9**, the presence of a hydrophobic benzyl group, as expected, resulted in a significant increase in the retention time compared to **A1** (Figure 2B). A new chiral center at the phosphorus atom to which the sulfonamide group is attached led to the separation of two diastereomers in both cases, more prominent in the case of the less-polar benzylamide **A9**. For most 15- and 17-mer 5′-unprotected oligonucleotides with one modification in different positions of the chain, a single peak corresponding to the main product was observed in the profiles with no detectable separation of diastereomers (Figure 2A, Appendix A). Yet, for some of the longer sequences, two peaks of stereoisomers were clearly distinguishable (Figure 3B, Appendix A).

The longer 17-mer oligonucleotide conjugates **A7**, **A8**, **A14**, and **A15** obtained with azide **A** with HBTU/HOBt activation (Appendix A), and the same sequences **B1**, **B2**, **B3**, and **B4** derived from pentafluorophenyl ester **B**, respectively (Figure 3B), all show a gradual increase in the retention time that correlates with the increase in the hydrophobicity of the amines from **1** to **4**. According to the elution profiles, the conjugation reaction with aqueous ammonia, *n*-propylamine, benzylamine, and 1-pyrenemethylamine proceeded with high yields in both series **A** and series **B**, with the latter resulting in cleaner products and slightly higher yields. In the case of azide **C**, only two derivatives were synthesized: amide **C1** and benzylamide **C2**. As evident from their HPLC profiles (Appendix A), these two conjugates contained higher amounts of impurities than in the case of **A** and **B**. This prompts us to conclude that 4-nitrophenyl 4-(azidosulfonyl)-benzoate **C** may also be used for oligonucleotide modification and conjugation, but with somewhat lesser efficiency.

The corresponding 15-mer oligonucleotide conjugates **D1**–**D4** obtained from pentafluorobenzyl ester **D** and with one or two *N*-(4-carboxybenzenesulfonyl)-phosphoramidate (ξ) groups were prepared in high yield (Appendix A) and used later to study the thermal stability of complementary duplexes with DNA and RNA (see below). The fully modified oligonucleotide conjugate **D6** with 1,1-dimethylethylenediamine residue at each internucleotidic position demonstrated a significant increase in retention time compared to the singly modified conjugate **D5** (Figure 4A). In turn, the singly modified conjugate **D5** showed a small decrease in the retention time compared to the unmodified control, presumably due to the presence of a polar zwitterionic group [83], but this effect was not observed for the 5′-dimethoxytritylated **D7** (Figure 4B). The separation of the main peak into the peaks of diastereomers was detected only for the fully modified **D6** and **D7**, which are the mixtures of 32 stereoisomers due to the presence of a chiral center at each of the five internucleotidic positions.

Additional characteristics of the modified oligonucleotides and conjugates were studied by 20% denaturing PAGE. Firstly, we compared the mobility of 15-mer oligodeoxyribonucleotides containing the carboxy or benzylamide residues in different positions of the oligonucleotide backbone. The mobility of oligonucleotides **A3**, **A4**, and **A5** containing carboxyl groups (Appendix A, lanes 2, 4, and 7) was close to the unmodified control oligonucleotide (Appendix A, lane 1), and, at the same time, was higher than the mobility of the conjugates **A11**, **A12**, and **A13** containing hydrophobic benzyl groups (Appendix A, lanes 3, 5, and 8). These data are in accord with previous observations that the amide-linked *N*-(4-carboxybenzenesulfonyl)-phosphoramidate group (ξ) in oligonucleotides, like in other sulfonyl phosphoramidate groups, is negatively charged under physiological pH, similar to the natural phosphodiester group [70,72].

Next, we studied the difference in electrophoretic mobility between the oligonucleotide **D5** with one zwitterionic group and a fully modified sequence **D6**, as well as their 5′-DMTr versions **D7** and **D8**. The presence of one zwitterionic group led to the decrease in the total negative charge of the oligonucleotide by one unit, which, in turn, resulted in a significant decrease in mobility (Figure 5, lanes 3 and 4) compared to the unmodified controls (Figure 5, lanes 1 and 2). The exhaustive substitution of all of the phosphate groups with zwitterionic groups dramatically reduced the mobility (Figure 5, lanes 7 and 8), but, nevertheless, did not result in the complete disappearance of the net negative charge of the oligonucleotide under the experimental conditions (pH 7.5). Interestingly, for the 5′-DMTr analog of the fully modified oligonucleotide, we observed practically no staining by the Stains-All dye (Figure 5A, lane 6), which may be explained by the screening of the remaining negative charge by the bulky DMTr group (compare Figure 5A,B). Other dyes need to be explored to ensure the efficient visualization of extensively modified oligonucleotide sequences on PAGE.

To ascertain the impact of the amide-linked *N*-(4-carboxybenzenesulfonyl)-phosphoramidate (ξ) group on the thermal stability of complementary duplexes with DNA and RNA, we prepared four 15-mer oligodeoxynucleotides **D1**, **D2**, **D3**, and **D4** containing a primary carboxamide modification (Figure 1, **3**, R′ = H), which had either one group at the 3′ or 5′-end or in the middle of the sequence, or two groups at both ends (Table 2). All of the oligonucleotides were analyzed by HPLC and purified by PAGE. UV melting studies revealed that one group at either end, or two of the modifications at both ends, did not have a significant adverse effect on the thermal stability of the duplexes of the modified oligonucleotides with either DNA or RNA in comparison with those of the unmodified control. The most destabilizing effect seemingly had the modification in the middle of the sequence (Table 2). Thus, one may conclude that the *N*-(4-carboxamidobenzenesulfonyl)-phosphoramidate (ξ) group may be well suited for a single modification at any position of an oligonucleotide chain, or for multiple modifications at the ends of the sequence.

## 3. Materials and Methods

### 3.1. Synthesis of Pentafluorophenyl, 4-Nitrophenyl, and Pentafluorobenzyl 4-(Azidosulfonyl)-benzoates

4-Carboxybenzenesulfonazide (**A**) was converted to either pentafluorophenyl (**B**) or 4-nitrophenyl (**C**) 4-(azidosulfonyl)-benzoate by reaction with the corresponding phenol promoted by dicyclohexylcarbodiimide (DCC) (see Appendix A for experimental details and characterization). Pentafluorobenzyl 4-(azidosulfonyl)-benzoate (**D**) was obtained by a two-step one-pot procedure from 4-(chlorosulfonyl)-benzoyl chloride by, firstly, a reaction with pentafluorobenzyl alcohol in the presence of 4-dimethylaminopyridine (DMAP) as a base and catalyst, and, secondly, by treatment with NaN_3_ in acetonitrile to give white crystalline **D** in good yield (see Appendix A for general information, experimental details, and spectra, see Appendix A).

### 3.2. Oligonucleotide Synthesis

Oligonucleotides were synthesized on an automated DNA/RNA ASM-800 synthesizer (Biosset, Novosibirsk, Russia) by the phosphoramidite method at the 0.2–0.4 μM scale using the corresponding deoxyribonucleotide 2-cyanoethyl-*N*,*N*-diisopropyl phosphoramidites and Controlled Pore Glass (CPG) 500 Å supports with attached deoxyribonucleosides (Sigma-Aldrich, Saint Louis, MO, USA). If a ξ modification was introduced manually, the preceding synthesis cycle was interrupted immediately before the oxidation step, the column with the support-bound 5′-DMTr-oligonucleotide was removed from the synthesizer, and the Staudinger reaction was carried out in the solid phase with 0.25 M solution of the corresponding sulfonyl azide **B**–**D** in acetonitrile for 30 min, or 1 h in the case of azide **A** in the presence of 5% *v*/*v* pyridine, to buffer the acidity of the free carboxyl group. After the reaction, the support was washed using 3 × 200 µL of CH_3_CN and dried in vacuo. If necessary, the column with the support-bound oligonucleotide was placed back into the synthesizer, and the synthesis was continued until completion. In the case of the automated insertion of the modification, a solution of the corresponding azide in acetonitrile (0.25 M) was placed into a synthesizer bottle and pumped through the column instead of iodine solution for the same time period. After the synthesis, the support-bound oligonucleotide was either treated with conc. aq. ammonia at 25 °C for 18 h to obtain the oligonucleotides with free carboxyl group **A1**–**A5** (Table 1), or transferred to the conjugation step. The conversion of the pentafluorobenzyl ester to amide in **D1**–**D4** was carried out after the Staudinger reaction by treatment of the polymer-bound oligonucleotides with conc. aq. ammonia solution at 55 °C for 18 h.

### 3.3. Synthesis of Oligonucleotide Conjugates

For the conjugation reaction, 60 µL of a solution of the desired amine **2**–**6** was added to the support-bound oligonucleotides. In the case of the oligonucleotides with *N*-(4-carboxybenzenesulfonyl)-phosphoramidate group (ξ) obtained from sulfonyl azide **A**, the carboxyl group was activated prior to amine addition by treatment with HBTU/HOBt (1:1 mol. ratio) at 35 °C in dry DMF for 35 min, as previously described [51]. The mixture was shaken for 1 h at 25 °C for all amines except **4**, which was continued for 2 h at 37 °C. *n*-Propylamine (**2**) and benzylamine (**3**) were used as 10% *v*/*v* in DMF; 1-(pyrenemethylamine (**4**) was used as 0.25 M solution in DMF–THF (1:1 *v*/*v*); 4,7,10-trioxa-1,13-tridecanediamine (**5**) and tetraethylenepentamine (**6**) were used as 5% *v*/*v* solution in DMF; 1,1-dimethylethylenediamine (**7**) was used neat, thereby combining the conjugation reaction, final deprotection of the protecting groups, and cleavage from solid support. After the completion of the conjugation reaction, the supernatant was removed, the support was washed by 3 × 200 µL of CH_3_CN, and dried in vacuo for 15 min. Finally, the protecting groups were removed and the conjugate was cleaved from the solid support by treatment with conc. aq. ammonia at 55 °C for 18 h for all conjugates except amine **7** (Table 1).

### 3.4. Reverse-Phase HPLC Analysis of Oligonucleotides and Conjugates

Analytical RP-HPLC was performed on a Milichrom A02 system (Econova Ltd., Chromatography Institute, Novosibirsk, Russia) with the use of a ProntoSIL 120-5 C18 AQ (2 mm × 75 mm, 5µ) column. The elution was carried out at a flow rate of 100 µL/min, and UV detection was conducted at 260 and 280 nm. Several elution gradients of acetonitrile in 20 mM TEAA, pH 7.0 were applied: from 0% to 50% in 30 min (i), from 0% to 60% in 30 min (ii), and from 0% to 50% in 30 min of 50% aq. acetonitrile in 20 mM TEAA buffer, pH 7.0 (iii).

### 3.5. Polyacrylamide Gel Electrophoresis under Denaturing Conditions

To control the homogeneity of the oligonucleotides, analytical electrophoresis was carried out in 0.4 mm thick 20% polyacrylamide gel (acrylamide—*N*,*N*′-methylene-*bis*-acrylamide (30:1)) at 50 V/cm voltage in 90 mM Tris–borate buffer, pH 8.3, containing 8 M urea and 2 mM Na_2_EDTA. Oligonucleotides were loaded onto the gel as solutions containing 8 M urea, 0.05% Xylene Cyanol FF, and 0.05% Bromophenol Blue. Bands were visualized by staining the gel with a solution of Stains-All dye (500 mg/L) in formamide followed by rinsing with distilled water.

Oligonucleotides synthesized for UV melting studies were isolated by preparative polyacrylamide gel electrophoresis in 2–3 mm thick 20% gel under denaturing conditions (8 M urea) and desalting on an NAP25 column with Sephadex G-25 (GE Healthcare, Buckinghamshire, UK) in the form of sodium salts.

### 3.6. UV Melting Studies

RNA template 5′-r(ATTTGAGCCTGGGAG) was kindly provided by Dr. Maria I. Meshchaninova. Thermal denaturation analysis of the duplexes of modified oligonucleotides with complementary DNA or RNA sequences was carried out in a quartz micro multi-cell cuvette with an optical path length of 1 mm using a UV-1800 UV–Vis spectrophotometer (Shimadzu, Kyoto, Japan) with thermoelectric cooling. All experiments were conducted in a buffer containing 10 mM Na-cacodylate, 5 mM MgCl_2_, and 100 mM NaCl, pH 7.4. The components were taken in stoichiometric ratios so that the total oligonucleotide duplex concentration in the buffer was 10 μM. The solutions were kept at 90 °C for 5 min, followed by a gradual decrease in the temperature from 90 °C to 15 °C to anneal the duplexes. Thereafter, the samples were heated from 15 °C to 90 °C at a rate of 0.2 °C min^−1^ to melt the duplex. Absorption spectra were recorded at 260 nm with a measurement step of 0.5 °C. Representative melting curves with DNA or RNA are included in Appendix A, respectively.

## 4. Conclusions

In this paper, we describe a convenient method for the conjugation of oligonucleotides via the amide bond formation at a modified phosphate position suitable for a range of amines including di- and polyamines that introduce positive charges into the oligonucleotide chain, which could be a way to increase cellular and tissue uptake and improve the bioavailability and activity of therapeutic oligonucleotides. Herein, we employed 4-carboxybenzenesulfonyl azide and its activated pentafluorophenyl, 4-nitrophenyl, or pentafluorobenzyl esters to obtain oligonucleotides carrying one or more internucleotidic *N*-(4-carboxybenzenesulfonyl)-phosphoramidate groups (ξ) via a modified phosphoramidite synthesis protocol substituting the Staudinger reaction for conventional aqueous iodine oxidation. The *N*-(4-carboxybenzenesulfonyl)-phosphoramidate group (ξ) is stable under the conditions of DNA synthesis. Its free carboxyl function does not require protection and it maintains a negative charge at physiological pH, similar to the native phosphodiester group. Thus, the carboxyl group can be introduced into any internucleotidic position of the sequence, whether 5′ or 3′-terminal, or internal, and, after suitable pre-activation by a peptide coupling agent such as HBTU/HOBt, it can undergo amide bond formation with a range of amines including those with positively charged side chains such as di- and polyamines. Although the activated aryl esters of the *N*-(4-carboxybenzenesulfonyl)-phosphoramidate proved to be too reactive to withstand the conditions of conventional DNA synthesis and, thus, allowed only the 5′-terminal conjugation to occur, the less-activated pentafluorobenzyl ester was sufficiently robust to be introduced at every phosphate position within the sequence, yet reactive enough to allow for smooth conversion to fully amide-modified oligonucleotides. The melting temperatures of the duplexes formed by ξ-oligodeoxynucleotides modified by one or two carboxamide side chains at the terminal or internal positions with complementary DNA or RNA were similar or only slightly lower than that of the unmodified DNA:DNA and DNA:RNA duplexes.

To conclude, the application of newly synthesized *N*-(4-carboxybenzenesulfonyl)-phosphoramidate-modified oligonucleotides either as activated esters or free carboxylic acids with subsequent HBTU/HOBt-mediated activation can be a useful way to prepare oligonucleotide conjugates with a range of suitable primary amine substrates. We are currently working on the extension of the approach described in this paper to oligonucleotide conjugates with peptides including cell-penetrating peptides known to greatly improve the cellular uptake of oligonucleotide therapeutics.

## Figures and Tables

**Figure 1 ijms-25-02007-f001:**
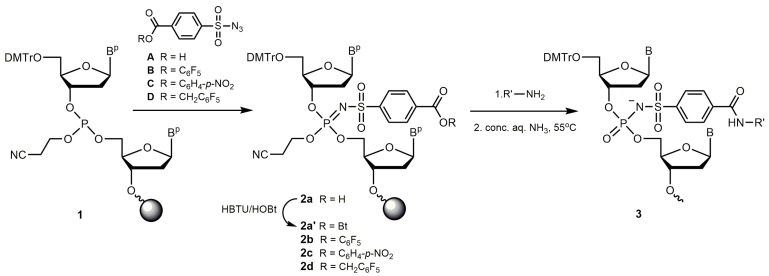
General scheme for the preparation of a carboxyl group containing phosphate-modified oligonucleotides and their amide-linked conjugates. Key: Bt—1*H*-benzotriazol-1-yl, DMTr—4,4′-dimethoxytrityl, HBTU—*O*-(1*H*-benzotriazol-1-yl)-*N*,*N*,*N*′,*N*′-tetramethyluronium hexafluorophosphate, HOBt—1-hydroxy-1*H*-benzotriazole; for R′, see Table 1.

**Figure 2 ijms-25-02007-f002:**
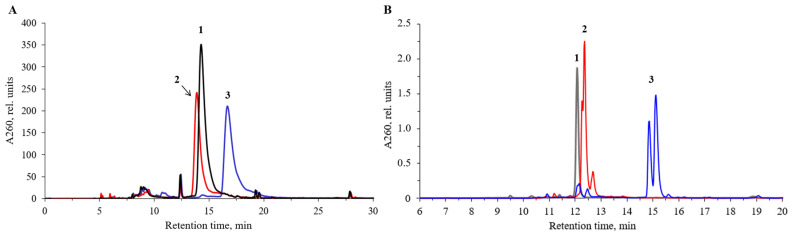
RP-HPLC elution profiles of crude oligonucleotides and conjugates: (**A**) unmodified oligonucleotide 5′-d(AGTCTCGACTTGCTACC) (**1**), **A6** (**2**), and conjugate **A14** (**3**); (**B**) unmodified oligonucleotide 5′-d(TTTTTT) (**1**), **A1** (**2**), and conjugate **A9** (**3**). Elution gradient (i) (Section 3).

**Figure 3 ijms-25-02007-f003:**
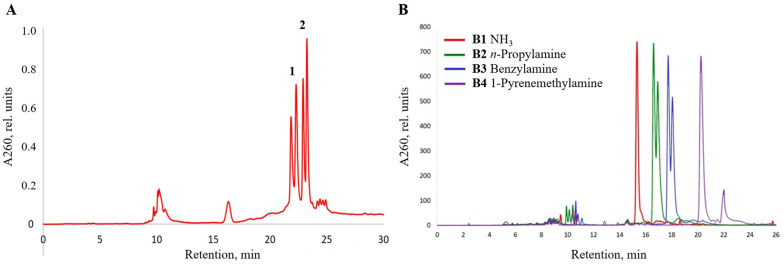
RP-HPLC elution profiles of crude reaction mixtures of (**A**) a co-injection of the carboxylic oligonucleotide **A3** (**1**) and the amide **D2** (**2**), elution gradient (iii); (**B**) conjugates **B1**, **B2**, **B3**, and **B4**, elution gradient (ii) (Section 3).

**Figure 4 ijms-25-02007-f004:**
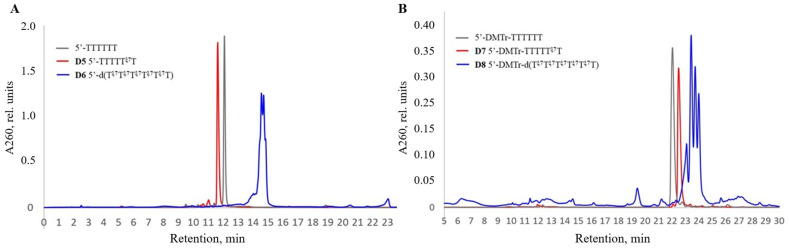
RP-HPLC profiles of crude oligonucleotide conjugate **D5**, **D6** (**A**) and **D7**, **D8** (**B**) compared to unmodified control 5′-d(TTTTTT) and 5′-DMTr-d(TTTTTT), respectively.

**Figure 5 ijms-25-02007-f005:**
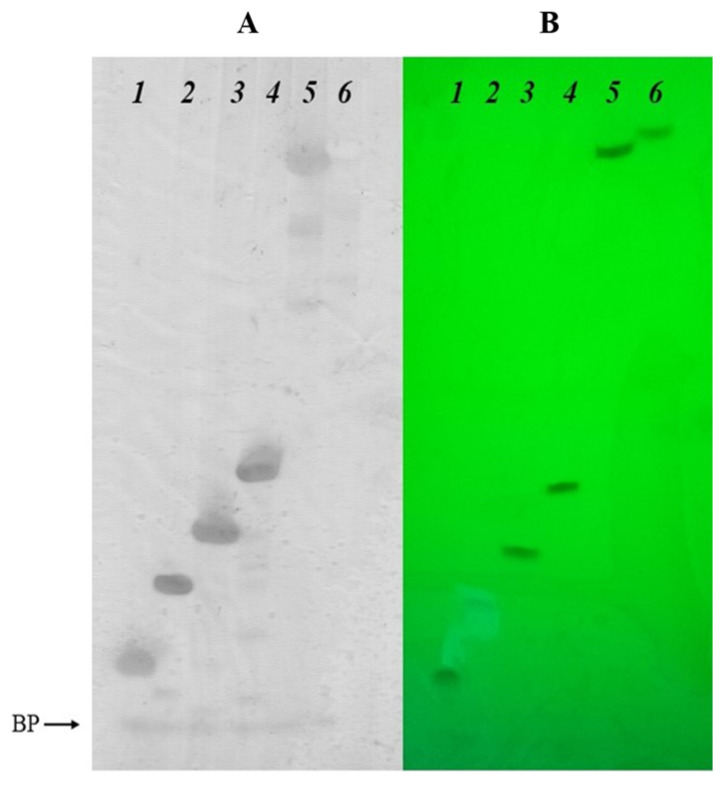
Electrophoretic comparison of the mobility of oligonucleotide conjugates with zwitterionic group: lanes: (*1*)—unmodified 5′-d(TTTTTT), (*2*)—unmodified 5′-DMTr-d(TTTTTT), (*3*)—sample **D5**, (*4*)—sample **D7**, (*5*)—sample **D6**, and (*6*)—sample **D8**. Visualized by staining with Stains-All dye (A) or UV shadowing (B).

**Table 1 ijms-25-02007-t001:** Amide-linked oligonucleotide conjugates with various primary amines obtained in this work.

No.	Primary Amine R′NH_2_	Oligonucleotide Sequence, 5′-3′ ^a^	Code ^b^	HPLC Conversion ^c^, %
**0**	Aqueous NH_3_ (25 °C) *Carboxylic acids*	T*^d^*TTTT^ξ^T	**A1**	74.8
		DMTr-TTTTT^ξ^T	**A2**	58.2
		C^ξ^TCCCAGGCTCAAAT	**A3**	71.4
		CTCCCAGGCTCAAA^ξ^T	**A4**	93.6
		CTCCCAGG^ξ^CTCAAAT	**A5**	93.7
		A^ξ^GTCTCGACTTGCTACC	**A6**	72.9
**1**	Aqueous NH_3_ (55 °C) R′ = –H *Amides*	A^ξ1^GTCTCGACTTGCTACC	**A7**	72.8
		A^ξ1^GTCTCGACTTGCTACC	**B1**	75.3
		A^ξ1^GTCTCGACTTGCTACC	**C1**	49.1
		CTCCCAGGCTCAAA^ξ1^T	**D1**	88.2
		C^ξ1^TCCCAGGCTCAAAT	**D2**	75.4
		CTCCCAGG^ξ1^CTCAAAT	**D3**	89.1
		C^ξ1^TCCCAGGCTCAAA^ξ1^T	**D4**	91.6
**2**	*n*-Propylamine R′ = –(CH_2_)_2_Me	A^ξ2^GTCTCGACTTGCTACC	**A8**	76.4
		A^ξ2^GTCTCGACTTGCTACC	**B2**	77.7
**3**	Benzylamine R′ = –CH_2_Ph	TTTTT^ξ3^T	**A9**	73.5
		DMTr-TTTTT^ξ3^T	**A10**	75.2
		C^ξ3^TCCCAGGCTCAAAT	**A11**	77.1
		CTCCCAGGCTCAAA^ξ3^T	**A12**	58.7
		CTCCCAGG^ξ3^CTCAAAT	**A13**	50.5
		A^ξ3^GTCTCGACTTGCTACC	**A14**	76.3
		A^ξ3^GTCTCGACTTGCTACC	**B3**	78.0
		A^ξ3^GTCTCGACTTGCTACC	**C2**	53.1
**4**	1-Pyrenemethylamine R′ = 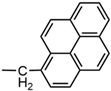	A^ξ4^GTCTCGACTTGCTACC	**A15**	65.7
		A^ξ4^GTCTCGACTTGCTACC	**B4**	67.2
**5**	4,7,10-Trioxa-1,13-tridecanediamine R′ = –(CH_2_)_3_[O(CH_2_)_2_]_2_O(CH_2_)_3_NH_2_	A^ξ5^GTCTCGACTTGCTACC	**A16**	45.6
		A^ξ5^GTCTCGACTTGCTACC	**B5**	60.3
**6**	Tetraethylenepentamine R′ = –(CH_2_)_2_NH[(CH_2_)_2_NH]_2_(CH_2_)_2_NH_2_	A^ξ4^GTCTCGACTTGCTACC	**A17**	52.7
		A^ξ6^GTCTCGACTTGCTACC	**B6**	57.5
**7**	1,1-Dimethylethylenediamine R′ = –(CH_2_)_2_NMe_2_	TTTTT^ξ7^T	**D5**	86.2
		T^ξ7^T^ξ7^T^ξ7^T^ξ7^T^ξ7^T	**D6**	87.9
		DMTr-TTTTT^ξ7^T	**D7**	78.1
		DMTr-T^ξ7^T^ξ7^T^ξ7^T^ξ7^T^ξ7^T	**D8**	66.3

^a^ The symbol (^ξ^) marks the positions of *N*-(4-carboxybenzenesulfonyl)-phosphoramidate groups, and the superscript numbers (**^1–7^**) next to the symbol refer to the corresponding conjugated amine. ^b^ In the codes, the letters indicate the corresponding sulfonyl azides **A**–**D** from which oligonucleotides or conjugates were obtained. Note: the phosphate modification is the same, but obtained using different azides **A**–**D** (Figure 1). ^c^ Calculated as the area of the peak of the product. ^d^ All oligonucleotides were oligodeoxynucleotides, prefix ‘d’ was omitted throughout. Color code: white—**A** only; gray—**A**/HBTU/HOBt; green—**B**; blue—**C**; beige—**D**.

**Table 2 ijms-25-02007-t002:** Thermal stability of the complementary duplexes of modified oligonucleotides with DNA and RNA.

Code	Oligonucleotide Sequence, 5′-3′	DNA Template 5′-d(ATTTGAGCCTGGGAG)		RNA Template 5′-r(ATTTGAGCCTGGGAG)	
		T_m_, °C	ΔT_m_*,* °C	T_m_, °C	ΔT_m_, °C
Control	d(CTCCCAGGCTCAAAT)	61.0 ± 0.4	–	65.7 ± 0.5	–
**D1**	d(CTCCCAGGCTCAAA^ξ1a^T)	61.2 ± 0.4	0.0 ± 0.4	65.6 ± 0.2	–0.1 ± 0.5
**D2**	d(C^ξ1^TCCCAGGCTCAAAT)	61.3 ± 0.2	–0.2 ± 0.5	65.0 ± 0.5	–0.7 ± 0.7
**D3**	d(CTCCCAGG^ξ1^CTCAAAT)	59.6 ± 0.4	–1.6 ± 0.6	64.3 ± 1.2	–1.4 ± 1.3
**D4**	d(C^ξ1^TCCCAGGCTCAAA^ξ1^T)	60.5 ± 0.5	–0.7 ± 0.6	65.1 ± 1.7	–0.6 ± 1.8

^a^ The symbol (^ξ1^) marks the position of *N*-(4-carboxamidobenzenesulfonyl)-phosphoramidate group (Table 1).

## Data Availability

Data is contained within the article and Appendix A.

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
