# Peer review of "A Convenient Oligonucleotide Conjugation via Tandem Staudinger Reaction and Amide Bond Formation at the Internucleotidic Phosphate Position"

_ijms, 2024, doi:10.3390/ijms25042007_

Round 1

Reviewer 1 Report

Comments and Suggestions for Authors

Stetsenko and coauthors described the Staudinger reaction of the phosphite triester during the synthetic cycle with various 4-carboxybenzenesulfonyl azides on a solid support during the oligonucleotide synthesis. It is an alternative means to reduce the charge of the oligonucleotides and has been successfully proved the survival of the synthetic cycles. The background was well introduced with the properly cited references. The chemistry part is done properly with plenty of analysis and characterization of experimental data. However, if the oligonucleotide synthesis detail parameters (coupling time, amidite conc. and equiv. etc) could be included in the SI, it would be useful for the new people to the oligonucleotide area. Also, hopefully, the biological experiment with various sequences will come soon. I look forward to seeing the improved cellular uptake results of the modified oligonucleotides.

 I recommend publishing without major modifications.

Author Response

  1. However, if the oligonucleotide synthesis detail parameters (coupling time, amidite conc. and equiv. etc) could be included in the SI, it would be useful for the new people to the oligonucleotide area.

            We thank the esteemed Reviewer for his/her encouraging and helpful comments. The information on synthetic cycle was included in the Supporting information as per Reviewer’s suggestion.

  1. Also, hopefully, the biological experiment with various sequences will come soon.

            We plan to extend this promising chemistry to the synthesis of biologically important antisense oligonucleotides as the next stage of our project.

  1. I look forward to seeing the improved cellular uptake results of the modified oligonucleotides.

            We do indeed hope to obtain therapeutic oligonucleotides with improved cellular uptake and tissue distribution by this method.

Reviewer 2 Report

Comments and Suggestions for Authors

This manuscript reports efficient method for the oligonucleotide conjugation via tandem of Staudinger reaction and amide bond formation at the internucleotidic phosphate linker. The authors employ an “unwanted by many oligonucleotides chemists” reaction of the tertiary phosphite linkers  generated during solid supported synthesis of oligo with azide group to their advantage by substituting conventional iodine oxidation with Staudinger reaction with simple azides. The methods describe here can be used to target other oligonucleotide conjugates including cell-penetrating peptides which could improve cellular uptake of oligonucleotide therapeutics. The final sulfonyl-phosphoramidate group maintains a negative charge at physiological PH like the native phosphodiesters. The manuscript is well-written, new methods/compounds are sufficiently characterized and relevant literature is adequately cited. The manuscript should be published as is.

Comments on the Quality of English Language

Minor English editing. For example, in conclusion line 406: substituting instead of substuting

Author Response

We would like to thank the esteemed Reviewer for his/her encouraging comments on our paper.

Minor English editing. For example, in conclusion line 406: substituting instead of substuting

            The point on English proofreading (line 406) was corrected in the updated version of the manuscript.

Reviewer 3 Report

Comments and Suggestions for Authors

The authors describe on a method for modifying the phosphate moiety of oligonucleotides on a solid phase based on the Staudinger reaction followed by the amidation. The authors' method is excellent, and has the potential to open up new chemical modifications of oligonucleotide therapeutics. This study is well performed and disserves publication in International Journal of Molecular Sciences. However, there are several points that I would like you to consider revising. In particular, I think additional data are needed regarding the evaluation of the physical properties of oligonucleotides.

1) The deprotection conditions for the cyanoethyl group mentioned on line 168 of the page 4 have been described in preveous literature as a method for suppressing cyanoethylation of nucleobases, so the following paper should be added as a reference.

Org. Proc. Res. Dev. 2003, 7, 832–838.

2) Please provide the values in Table 1 that represent the amidation efficiency. (I think the yield of the final product is also fine.) Although the amidation efficiency is clearly explained, it would be easier to understand if the yield is listed in Table 1.

3) Is "A3A5" on page 8, line 276 correct?  Oligonucleotides "A3A5" are carboxylic acids, but the sentence on line 274–276 states "15-mer oligonucleotides containing amide and benzylamide residues".

4) As mentioned in the introduction, cationic modification is an effective approach to improve the properties of oligonucleotide therapeutics. From this point of view, for oligonucleotides No.5 and No.6, sequences with amine-modification at different positions, like D1D4, should be prepared, and physical properties (PAGE and UV-melting study) should be evaluated.

5) For other backbone modifications, duplex stability often differs slightly between diastereomers. The oligonucleotide used in this study is a diastereomer mixture. Is the Tm the same between each diastereomer? It is interesting to see if a simple sigmoid curve is obtained. Please add representative melting data to the supporting information. 

6) Empirically, Tm values are rarely completely reproducible, and Tm values include a certain amount of measurement error. Are the Tm values in Table 2 the average values after several measurements? Please write in the experimental section how much measurement error (±X °C) the Tm values in Table 2 includes. 

Also, although it depends on the extent of the measurement error, if the decrease in the Tm value of D3 is within the measurement error range, it cannot be concluded that "The most destabilizing effect had the modification in the middle of the sequence D3, which exceeded that of the doubly modified oligonucleotide D4 (Table 2)."

7) Some of the 13C NMR spectra in the supporting information are out of phase, so please replace them with spectra that have been reanalyzed. 

8) Please add "NMR" after "13C" on page 6, line 21 in the supporting information.

Author Response

1) The deprotection conditions for the cyanoethyl group mentioned on line 168 of the page 4 have been described in preveous literature as a method for suppressing cyanoethylation of nucleobases, so the following paper should be added as a reference.

Org. Proc. Res. Dev. 2003, 7, 832–838.

            We thank the esteemed Reviewer for his/her helpful suggestions. The mentioned reference was added to the list as ref. [81].

2) Please provide the values in Table 1 that represent the amidation efficiency. (I think the yield of the final product is also fine.) Although the amidation efficiency is clearly explained, it would be easier to understand if the yield is listed in Table 1.

            Table 1 was expanded to include the HPLC conversion yields calculated as percentage area of the product peak.

3) Is "A3A5" on page 8, line 276 correct?  Oligonucleotides "A3A5" are carboxylic acids, but the sentence on line 274–276 states "15-mer oligonucleotides containing amide and benzylamide residues".

            Thanks for spotting this out. The lines 274-276 were corrected to read “the carboxyl group” instead of “amide” as in Figure S11.

4) As mentioned in the introduction, cationic modification is an effective approach to improve the properties of oligonucleotide therapeutics. From this point of view, for oligonucleotides No.5 and No.6, sequences with amine-modification at different positions, like D1D4, should be prepared, and physical properties (PAGE and UV-melting study) should be evaluated.

            We are currently exploring a range of cationic side-chains in zwitter-ionic oligonucleotide context in order to elucidate possible structure-activity relationships (see, e.g. Patrushev et al, Mol. Biol. 2023, 57, 320). However, this work is far from complete yet. Therefore, we would rather prefer to publish it as a systematic study when more data is available.

5) For other backbone modifications, duplex stability often differs slightly between diastereomers. The oligonucleotide used in this study is a diastereomer mixture. Is the Tm the same between each diastereomer? It is interesting to see if a simple sigmoid curve is obtained. Please add representative melting data to the supporting information. 

            The melting curves were added to the Supporting Material as Figures S23-S24. As no attempt to separate the diastereomers was made in this study, we have no data yet to publish on the thermal denaturation of pure diastereomers although we anticipate their different binding affinity. However, we do plan to ascertain the single diastereomer RNA binding differences in near future, albeit in a slightly different context.

6) Empirically, Tm values are rarely completely reproducible, and Tm values include a certain amount of measurement error. Are the Tm values in Table 2 the average values after several measurements? Please write in the experimental section how much measurement error (±X °C) the Tm values in Table 2 includes. 

Also, although it depends on the extent of the measurement error, if the decrease in the Tm value of D3 is within the measurement error range, it cannot be concluded that "The most destabilizing effect had the modification in the middle of the sequence D3, which exceeded that of the doubly modified oligonucleotide D4 (Table 2)."

            Thanks for expanding on this important point. We updated Table 2 to include the error calculated from at least triplicate experiments. In the case of RNA duplex it was indeed significant enough to prompt us to correct the statement picked up by the Reviewer (lines 310-311).

7) Some of the 13C NMR spectra in the supporting information are out of phase, so please replace them with spectra that have been reanalyzed. 

            The 13C NMR spectra in Figures S14, S15, and S18 in the Supporting Material are the JMOD spectra, in which the signals from the carbon atoms with even number of hydrogens (0, 2) are shown in the upper half of the spectrum, and the signal corresponding to the carbons with the odd number of hydrogens (1, 3) are shown in the lower half of the spectrum. In the updated version of the Supporting Material, the 13C NMR spectrum in Figure S21 was also replaced with the JMOD spectrum for consistency.

8) Please add "NMR" after "13C" on page 6, line 21 in the supporting information.

            Added.

Round 2

Reviewer 3 Report

Comments and Suggestions for Authors

The manuscript has been revised well. I think this manuscript will be acceptable.